# Utilizing Multi-Dimensional MmWave MIMO Channel Features for Location Verification

**DOI:** 10.3390/s22239202

**Published:** 2022-11-26

**Authors:** Pinchang Zhang, Yangyang Liu, Ji He

**Affiliations:** 1School of Computer Science, Nanjing University of Posts and Telecommunications, Nanjing 210023, China; 2School of Computer Science, Xidian University, Xi’an 710071, China

**Keywords:** location verification, physical layer authentication, millimeter-wave (mmWave) communications, multiple-input multiple-output (MIMO), channel features

## Abstract

In this paper, we address the problem of authenticating transmitters in millimeter-wave (mmWave) multiple-input multiple-output (MIMO) communication systems, and propose a location verification scheme based on multi-dimensional mmWave MIMO channel features. In particular, we first examine the mmWave MIMO channel features in terms of azimuth angle of arrival (AAoA), elevation angle of arrival (EAoA), and path gain, and then extract these fine-grained channel features through the maximum-likelihood (ML) estimation method. Based on the extracted feature parameters, authentication validation is cast in the framework of hypothesis testing theory. We also derive the analytical expressions for the typical false alarm and detection rates by using the likelihood ratio test and thus the statistical performance is analytically established. Finally, extensive numerical results are provided to demonstrate the performance of the proposed authentication scheme.

## 1. Introduction

Millimeter-wave (mmWave) multiple-input multiple-output (MIMO) systems serve as a critical technique for 5G (fifth-generation communication) and beyond wireless communication networks. The mmWave MIMO technique can lead to a significant increase in data rates, wide bandwidth, and higher spectrum efficiency. By integrating huge antenna arrays, 5G and beyond communication systems operating from 30 to 300 GHz can support huge connections among a bulk of smart devices, contributing to various applications such as the Industrial Internet of Things (IIoT) and the Internet of Vehicles (IoV).

However, mmWave MIMO communication systems may become subject to malicious attacks because of the above important applications they can support. Moreover, the mmWave wireless communication medium is open in space and thus attackers can easily access the communication network and obtain the transferred information between legitimate users and base stations. One of the main threats in mmWave MIMO systems is identity-based spoofing attack, where an attacker eavesdrops the legitimate communication links and further impersonates a legitimate transmitter to inject harmful signals into the communication network. To tackle the spoofing attacks in mmWave systems, identity authentication presents a promising solution to secure mmWave MIMO communication systems.

We note that the existing authentication schemes can be roughly divided into higher layer and physical layer authentication schemes. The higher layer authentication schemes are mainly dependent on secret keys and encryption/decryption algorithms to validate transmitter identities [1,2,3]. Meanwhile, the physical layer authentication schemes verify transmitter identities through physical layer features [4]. It is noted that the higher layer authentication may not completely adapt to the emerging mmWave MIMO systems which include distributed, dynamic and highly heterogeneous devices. Such mmWave MIMO systems may not support the cryptographic key distribution and management, high consumption of communication and computation resources in higher layer authentication schemes. As a complement and an enhancement of the higher layer authentication, physical layer authentication possesses several characteristics such as high security, low complexity and high compatibility [5]. Thus, in this paper we focus on physical layer authentication in mmWave MIMO systems.

Existing physical layer authentication schemes mainly include channel-based authentication and hardware-impairment-based authentication. The hardware-impairment-based method authenticates the legitimacy of a transmitter identity by extracting specific non-ideal hardware parameters from received signals. In contrast, channel-based authentication utilizes the fact that wireless channels decorrelate rapidly as the source (e.g., a transmitter or user using transmitter is transmitting signals to an intended destination/receiver) changes location in an environment with rich scatterers and reflectors. In fact, channel-based authentication verifies the location of the current transmitter rather than transmitter physics identity, and hence is referred to as location verification. There exists some research works on physical layer authentication for mmWave systems [6,7,8,9,10,11,12,13,14,15]. In [6], the authors propose a physical layer authentication scheme by using sparse mmWave channel. A power allocation technique for the precoder and the authentication tag is first developed for fingerprint-based authentication in a multi-user mmWave UAV network [7]. Multipath time delay caused by planar specular reflectors is acquired to illustrate the authenticity of user position 5G mmWave communication with planar reflectors [8]. The intrinsic physical attributes for 5G communications are analyzed for physical layer authentication in detail and the future possible directions for the exploitation of physical attributes is demonstrated in [9]. The authors in [10] further study the physical characteristics of 5G IoT in various application scenarios and describe the possible security threats resulting from the 5G IoT physical layer. Based on virtual angles of arrival in mmWave systems and machine learning methods, the authors in [11] propose a physical layer authentication scheme to detection spoofing attacks. Two effective pilot contamination attack detection approaches are proposed for power non-orthogonal multiple access in massive MIMO 5G communications, based on the sparseness and statistics of mmWave virtual channels [12]. Following this line, the authors in [13] present a new physical layer countermeasure using channel virtual representation to defend against pilot contamination attack. In [14], unique beam pattern features are utilized to achieve physical layer spoofing attack detection and the detection problem is cast as a machine learning classification through the sector level sweep (SLS) process. The authors in [15] develop a physical layer authentication scheme by using signal-to-noise ratio (SNR) trace features to detect spoofing attacks in mmWave systems.

Frequency-selective Rayleigh channel time variations are used to detect spoofing attacks in wireless networks, considering the channel estimation errors [16]. To improve the spoofing detection accuracy, a logistic regression-based authentication approach is developed without the requirements of the known channel model [17]. The authors jointly use multipath delay and channel gain characteristics of the channel impulse response to achieve performance improvement of physical layer authentication [18]. The above existing approaches demonstrate the uniqueness and usability of physical layer features for spoofing attack detection and authentication. It is noted that the above existing approaches are developed for spoofing detection by conventional channel state information. These approaches cannot be extended directly to mmWave MIMO communication systems because it is difficult for them to sufficiently excavate fine-grained physical layer characteristics from mmWave MIMO channels. Hardware-impairment-based authentication generally needs to employ high-precision hardware equipment for hardware feature extraction and bears high computational complexity [14,15]. Moreover, machine-learning-based authentication framework requires an expensive training stage with a large amount of training data, which might be unrealistic in practice. In addition, the analytic establishment of statistical performances still remains an open problem in machine-learning-based authentication frameworks. To this end, we explore the exploitation of multi-dimensional mmWave MIMO channel features to design a new location verification scheme for mmWave MIMO communication systems. These channel features include azimuth angle of arrival (AAoA), elevation angle of arrival (EAoA) and path gain, which are jointly used to validate the legitimacy of transmitters and to improve the authentication performance. The main difference between the schemes proposed in this paper and those in previous works [16,17,18] is that the previous approaches mainly focus on channel statistics in the time domain while this paper extracts channel features in the time domain as well as the angle domain. In addition, the scheme proposed in this paper attempts to explore the exploitation of multi-dimensional mmWave MIMO channel features to achieve location verification.

The main contributions of this paper are summarized as follows:By exploiting multi-dimensional mmWave MIMO channel features in both the time domain and angle domain, we develop a new location verification scheme to achieve validation of a transmitter location for mmWave MIMO communication systems.To determine estimation error variances of channel features in terms of AAoA, EAoA, and path gain, we estimate the mmWave MIMO channels based on maximum-likelihood estimation theory. To analytically evaluate the performance of our proposed location verification scheme, we derive the typical two performance metrics and the statistical performance is analytically established.Extensive numerical results are provided to demonstrate that the proposed scheme achieves desired performance. Numerical results are used to further show how the system parameters can affect the statistical performance.

The remainder of this paper is organized as follows. Section 2 introduces the concerned system model and problem formulation. Section 3 and Section 4 present the proposed location verification scheme and performance analysis, respectively. Section 5 shows the numerical results and performance evaluation. Finally, Section 6 concludes this paper.

Notation: ${\left(\cdot \right)}^{H}$, ${\left(\cdot \right)}^{T}$, ${\left(\cdot \right)}^{*}$, ${\left(\cdot \right)}^{-1}$, ${\left(\cdot \right)}^{†}$ denote the conjugate-transpose, transpose, conjugate, inverse and pseudo-inverse of a matrix, respectively. The (m,n)-th element of a matrix H is $H_{m,n}$. $\mathrm{diag}(a_{1},...,a_{N})$ is a diagonal matrix with $a_{1},...,a_{N}$ on the main diagonal. vec(H) is the vectorization of matrix H. tr(·) and ≜ represent the matrix trace function and definitions, respectively. E{·} represents the expectation operator. $\left|\right|\cdot {\left|\right|}_{2}$ is the Euclidean norm operator. A Gaussian random variable *x* with mean μ and variance *c* can be represented by x∼N(μ,c). If let 〈X〉 be the subspace spanned by the columns of matrix X, then $P_{X}$ denotes the orthogonal projection matrix onto 〈X〉, that is, $P_{X}=X{\left(X^{H}X\right)}^{-1}X^{H}.$

## 2. Problem Formulation and System Model

### 2.1. Problem Formulation

Similar to previous works [16,17,18], we also use a three-entity model including a legitimate transmitter (Alice), a malicious attacker (Eve) and a legitimate base station (Bob), where Alice and Eve are both equipped with a single antenna and Bob is equipped with $N_{r}$ antennas, as shown in Figure 1. Eve intends to spoof Bob by using the identity of Alice and thus may launch some more advanced attacks, such as a man-in-the-middle attack and session hijacking attack while Alice communicates with Bob.

Suppose Alice transmits one data frame to Bob at time t−1 and this data frame was validated by cryptography-based higher layer authentication approaches. In this process, Bob measures and stores physical layer features information, e.g., mmWave MIMO channel features, with the help of signal estimation techniques. Subsequently, at time *t*, an unknown transmitter (either Alice or Eve) sends another data frame to Bob and Bob needs to validate the data frame origin (i.e., the current transmitter identity) to avoid impersonation attack.

Eve has data frame structure information by probing the transmitted signals between Alice and Bob and some publicly known information (e.g., training sequences and pilot signals). The estimation techniques and authentication schemes that Bob utilizes cannot be known by the attacker, Eve. Further, we exploit the carrier sense multiple access/collision detection protocol.

### 2.2. Channel Model

Bob has a $N_{r}$-antenna uniform planar array (UPA) where the numbers of antennas in the horizon and vertical directions are *M* and *N*, respectively, where $N_{r}=MN$. The number of radio frequency chains of Bob is $M_{RF}$ [19]. Let *d* be the distance between any arbitrary two adjacent antennas, and $G_{RF}$ be the analog beamformer equipped at Bob. Considering the limited scattering characteristics of the mmWave communication environments, we here adopt the extended Saleh-Valenzuela (SV) model to characterize the mmWave MIMO channels [20,21]. Denote $\phi_{l}$ as the azimuth angle of arrival (AAoA) of the *l*-th propagation multipath of the transmitter and $\theta_{l}$ as the elevation angle of arrival (EAoA) of that, for l=1,2,⋯,L, respectively. We assume $\phi_{l}∼N\left(0,\sigma_{\phi_{l,XB}}^{2}\right)$ and $\theta_{l}∼N\left(0,\sigma_{\theta_{l,XB}}^{2}\right)$. The steering matrix $A(\phi_{l},\theta_{l})$ is expressed by
(1)$$\begin{array}{c} A\left(\phi_{l},\theta_{l}\right)=\frac{1}{\sqrt{MN}}\left[\begin{array}{ccc} 1 & \cdots & e^{j2\pi d/\lambda \left(N-1\right)\sin\phi_{l}\cos\theta_{l}}\\ \vdots & \ddots & \vdots \\ e^{j2\pi d/\lambda \left(M-1\right)\cos\phi_{l}} & \cdots & e^{j2\pi d/\lambda (\left(M-1\right)\cos\phi_{l}+\left(N-1\right)\sin\phi_{l}\cos\theta_{l})} \end{array}\right], \end{array}$$
where λ is the wavelength of the carrier signal [22]. For simplicity, let $v_{1,l}=\frac{2\pi d}{\lambda }\cos\phi_{l}$ and $v_{2,l}=\frac{2\pi d}{\lambda }\sin\phi_{l}\cos\theta_{l}$. The steering matrix $A(\phi_{l},\theta_{l})$ can be further written as
(2)$$\begin{array}{cc} A(\phi_{l},\theta_{l}) & =A\left(v_{1,l},v_{2,l}\right)=a\left(v_{1,l}\right)a^{T}\left(v_{2,l}\right), \end{array}$$
where $a\left(v_{1,l}\right)=\frac{1}{\sqrt{M}}{\left[1,\cdots ,e^{j\left(M-1\right)v_{1,l}}\right]}^{T}$ and $a\left(v_{2,l}\right)=\frac{1}{\sqrt{N}}{\left[1,\cdots ,e^{j\left(N-1\right)v_{2,l}}\right]}^{T}$.

Similar to [23], the mmWave channel with *L* being the number of propagation paths is modeled as
(3)$$\begin{array}{cc} H & =\frac{1}{\sqrt{L}}\sum_{l=1}^{L}\alpha_{l}A\left(\phi_{l},\theta_{l}\right)=\frac{1}{\sqrt{L}}\sum_{l=1}^{L}\alpha_{l}a\left(v_{1,l}\right)a^{T}\left(v_{2,l}\right)=\frac{1}{\sqrt{L}}A_{v_{1}}H_{\alpha }A_{v_{2}}^{T}, \end{array}$$
where $A_{v_{1}}=\left[a\left(v_{1,1}\right),a\left(v_{v_{1,2}}\right),\cdots ,a\left(v_{1,L}\right)\right]$, $A_{v_{2}}=\left[a\left(v_{2,1}\right),a\left(v_{v_{2,2}}\right),\cdots ,a\left(v_{2,L}\right)\right]$, $H_{\alpha }=\mathrm{diag}\left(\alpha_{1},\alpha_{2},\cdots ,\alpha_{L}\right)$ and $\alpha_{l}∼\mathrm{CN}\left(0,\sigma_{\alpha_{l,XB}}^{2}\right)$ is the path gain of the *l*-th propagation path [22,24,25].

The (m,n)-th element (m=0,1,⋯,M−1 and n=0,1,⋯,N−1) for matrix H is
(4)$$\begin{array}{c} H_{m,n}=\frac{1}{\sqrt{LMN}}\sum_{l=1}^{L}\alpha_{l}e^{j(mv_{1,l}+nv_{2,l})}. \end{array}$$

Path gain is assumed to keep unchanged during one frame [26]. Based on [27], we use first-oder Gauss–Markov process to model the time-varying path gain as
(5)$$\begin{array}{c} \alpha_{l}\left(t\right)=\rho_{\alpha }\alpha_{l}\left(t-1\right)+\sqrt{1-\rho_{\alpha }^{2}}u_{\alpha_{l}}\left(t\right), \end{array}$$
where $\rho_{\alpha }$ is the path gain correlation coefficient [28] and $u_{\alpha_{l}}\left(t\right)$ is zero-mean Gaussian noise, i.e., $u_{\alpha_{l}}\left(t\right)∼\mathrm{CN}\left(0,\sigma_{\alpha_{l,XB}}^{2}\right)$.

Similarly, we use first-order Gauss–Markov process to model the time-varying azimuth angle of arrival as
(6)$$\begin{array}{c} \phi_{l}\left(t\right)=\rho_{\phi }\phi_{l}\left(t-1\right)+\sqrt{1-\rho_{\phi }^{2}}u_{\phi_{l}}\left(t\right), \end{array}$$
where $\rho_{\phi }$ is the AAoA correlation coefficient [28] and $u_{\phi_{l}}\left(t\right)$ is zero-mean Gaussian noise, i.e., $u_{\phi_{l}}\left(t\right)∼N\left(0,\sigma_{\phi_{l,XB}}^{2}\right)$. We also use first-order Gauss–Markov process to model the time-varying elevation angle of arrival as
(7)$$\begin{array}{c} \theta_{l}\left(t\right)=\rho_{\theta }\theta_{l}\left(t-1\right)+\sqrt{1-\rho_{\theta }^{2}}u_{\theta_{l}}\left(t\right), \end{array}$$
where $\rho_{\theta }$ is the EAoA correlation coefficient [28] and $u_{\theta_{l}}\left(t\right)$ is zero-mean Gaussian noise, i.e., $u_{\theta_{l}}\left(t\right)∼N\left(0,\sigma_{\theta_{l,XB}}^{2}\right)$.

Channel-based verification utilizes the fact that wireless channels decorrelate rapidly as the source (e.g., a transmitter or user using transmitter is transmitting signals to an intended destination/receiver) changes location in an environment with rich scatterers and reflectors. This indicates that wireless channel features are spatially decorrelated between different geographic locations. When the distance between a legitimate source and an adversary is greater than half of the wavelength, it is extremely difficult for the adversary to obtain the accurate wireless channel features. Although the channel-based method essentially authenticates the practical geographic location related to the inherent properties of channel features, it is an effective authentication to achieve legitimacy validation of the source identity. Therefore, this paper aims to design a new location verification scheme for validating data frame origin to resist against the impersonation attack.

### 2.3. Communication Model

The baseband signal received by Bob at time *t* is written as
(8)$$\begin{array}{c} r\left(t\right)=G_{RF}^{H}\mathrm{vec}\left(H_{XB}\left(t\right)\right)s\left(t\right)+n\left(t\right), \end{array}$$
where

s(t) is the transmitted signal at time *t*. The signal power is $p=E\{|s\left(t\right){|}^{2}\}$.$H_{XB}\left(t\right)$ denotes the channel matrix between transmitter *X* (either Alice or Eve) and receiver Bob.n(t) is a zero-mean complex additive white Gaussian noise (AWGN), i.e., $n\left(t\right)∼\mathrm{CN}\left(0,\sigma_{n}^{2}I\right)$.

## 3. Proposed Location Verification Scheme

### 3.1. Estimation of MmWave Channels

To exploit the fine-grained mmWave channel features, we estimate the channel parameters separately. We first estimate AAoA and EAoA, and then estimate the mmWave path gain. We can write the estimations of $\phi_{l,XB}\left(t\right)$, $\theta_{l,XB}\left(t\right)$ and $\alpha_{l,XB}\left(t\right)$, respectively, as
(9)$$\begin{array}{c} {\hat{\phi }}_{l,XB}\left(t\right)=\phi_{l,XB}\left(t\right)+\Delta \phi_{l}\left(t\right), \end{array}$$
(10)$$\begin{array}{c} {\hat{\theta }}_{l,XB}\left(t\right)=\theta_{l,XB}\left(t\right)+\Delta \theta_{l}\left(t\right), \end{array}$$
(11)$$\begin{array}{c} {\hat{\alpha }}_{l,XB}\left(t\right)=\alpha_{l,XB}\left(t\right)+\Delta \alpha_{l}\left(t\right), \end{array}$$
where $\Delta \phi ∼N(0,\sigma_{\Delta \phi }^{2})$, $\Delta \theta ∼N(0,\sigma_{\Delta \theta }^{2})$ and $\Delta \alpha ∼\mathrm{CN}(0,\sigma_{\Delta \alpha }^{2})$ are the corresponding estimation errors, respectively. Next, we calculate the variances of the estimation errors, i.e., $\sigma_{\Delta \phi }^{2}$, $\sigma_{\Delta \theta }^{2}$ and $\sigma_{\Delta \alpha }^{2}$.

For notational simplicity, here we ignore the time index *t* and the path index *l*. The signal received by Bob at time *t* is given by
(12)$$\begin{array}{cc} r & =G_{RF}^{H}\mathrm{vec}\left(H\right)s+n=G_{RF}^{H}\mathrm{vec}\left(A\right)\alpha s+n, \end{array}$$
where A is the M×N steering matrix with its (m,n)-th element given by
(13)$$\begin{array}{c} A_{m,n}=\frac{1}{\sqrt{MN}}e^{j(mv_{1}+nv_{2})}. \end{array}$$

The conditional PDF of r is expressed by
(14)$$\begin{array}{c} f\left(r|v_{1},v_{2},\alpha \right)=\frac{1}{{\left(\pi \sigma_{n}^{2}\right)}^{M_{RF}}}\exp\left\{-\frac{||r-G_{RF}^{H}\mathrm{vec}\left(A\right)\alpha s{\left|\right|}^{2}}{\sigma_{n}^{2}}\right\}. \end{array}$$

The maximum-likelihood (ML) method in [29] is used to estimate mmWave channel features $v_{1}$, $v_{2}$, and α. The estimation of these features can be expressed as
(15)$$\begin{array}{c} \left[\hat{v_{1}},\hat{v_{2}},\hat{\alpha }\right]=\underset{v_{1},v_{2},\alpha }{\arg\max}f\left(r|v_{1},v_{2},\alpha \right). \end{array}$$

Based on (Equation 14) and (Equation 15), we can equivalently write (Equation 15) as
(16)$$\begin{array}{c} \left[\hat{v_{1}},\hat{v_{2}},\hat{\alpha }\right]=\underset{v_{1},v_{2},\alpha }{\arg\min}||r-G_{RF}^{H}\mathrm{vec}\left(A\right)\alpha s{\left|\right|}^{2}. \end{array}$$

When giving $v_{1}$ and $v_{2}$, the estimation of α is calculated as
(17)$$\begin{array}{c} \hat{\alpha }=\mathrm{vec}{\left(A\right)}^{H}{\left(G_{RF}^{H}\right)}^{†}s^{*}r. \end{array}$$

Substituting (Equation 17) into (Equation 16), the estimations of $v_{1}$ and $v_{2}$ is written as
(18)$$\begin{array}{cc} {}[\hat{v_{1}},\hat{v_{2}}]\; & =\underset{v_{1},v_{2}}{\arg\min}||r-G_{RF}^{H}\mathrm{vec}\left(A\right)\mathrm{vec}{\left(A\right)}^{H}{\left(G_{RF}^{H}\right)}^{†}s^{*}rs{\left|\right|}^{2}\\ \; & ≜\underset{v_{1},v_{2}}{\arg\max}r^{H}G_{RF}^{H}P_{b}{\left(G_{RF}^{H}\right)}^{†}r=\underset{v_{1},v_{2}}{\arg\max}g\left(v_{1},v_{2}\right), \end{array}$$
where $g(v_{1},v_{2})$ is the cost function related to $v_{1}$ and $v_{2}$, $\sigma_{s}^{2}=s^{*}s$ is the signal power and $P_{b}=\mathrm{vec}\left(A\right)\mathrm{vec}{\left(A\right)}^{H}$.

Next, we give Lemma 1 regarding the variance of the estimations for $v_{1}$ and $v_{2}$.

**Lemma 1.** 
*The variance of the estimations for $v_{1}$ and $v_{2}$ in (Equation 18) are given by*

(19)
$$\begin{array}{c} c_{\Delta v_{k}}=\frac{\sigma_{n}^{2}}{2\sigma_{s}^{2}{\left|\alpha \right|}^{2}\mathrm{vec}{\left(A\right)}^{H}V_{k}P_{b}^{⊥}V_{k}\mathrm{vec}\left(A\right)},\;\;k=1,2, \end{array}$$

*where $P_{b}^{⊥}=I-P_{b}$. $V_{1}$ and $V_{2}$ are the diagonal matrices given by*

(20)
$$\begin{array}{c} V_{1}=\mathrm{diag}\left(\underset{N}{\underset{︸}{0.\cdots ,0}},\cdots ,\underset{N}{\underset{︸}{\left(M-1),\cdots ,(M-1\right)}}\right), \end{array}$$


(21)
$$\begin{array}{c} V_{2}=\mathrm{diag}\left(\underset{M}{\underbrace{0.\cdots ,0}},\cdots ,\underset{M}{\underbrace{\left(N-1),\cdots ,(N-1\right)}}\right). \end{array}$$



**Proof.** The received signal without noise can be written as
(22)$$\begin{array}{c} r_{s}=G_{RF}^{H}\mathrm{vec}\left(A\right)\alpha s. \end{array}$$When considering the receiver noise, the received signal can be further written as $r=r_{s}+n$. By using Taylor series expansion, the first derivative of the cost function $g(v_{1},v_{2})$ is given by
(23)$$\begin{array}{cc} \; & \frac{\partial g(v_{1},v_{2})}{\partial v_{k}}{|}_{v_{e}={\hat{v}}_{k}}\approx \frac{\partial g(v_{1},v_{2})}{\partial v_{k}}{{|}_{v_{e}={\hat{v}}_{k}}+\frac{\partial^{2}g\left(v_{1},v_{2}\right)}{\partial^{2}v_{k}}|}_{v_{e}={\hat{v}}_{k}}\Delta v_{k}, \end{array}$$
where $\Delta v_{k}={\hat{v}}_{k}-v_{k}$ denotes perturbation on $v_{k}$.Setting (Equation 23) equal to zero, $\Delta v_{k}$ is given by
(24)$$\begin{array}{c} \Delta v_{k}=-\frac{\frac{\partial g(v_{1},v_{2})}{\partial v_{k}}{|}_{v_{e}={\hat{v}}_{k}}}{\frac{\partial^{2}g\left(v_{1},v_{2}\right)}{\partial^{2}v_{k}}{|}_{v_{e}={\hat{v}}_{k}}}=-\frac{\dot{g}\left(v_{1},v_{2}|v_{k}\right)}{\ddot{g}\left(v_{1},v_{2}|v_{k}\right)}, \end{array}$$
where $\dot{g}\left(v_{1},v_{2}|v_{k}\right)$ and $\ddot{g}\left(v_{1},v_{2}|v_{k}\right)$ are the first- and second-order derivatives of the cost function $g(v_{1},v_{2})$ conditioned on $v_{k}$, respectively.$\dot{g}\left(v_{1},v_{2}|v_{k}\right)$ is expressed as
(25)$$\begin{array}{cc} \dot{g}\left(v_{1},v_{2}|v_{k}\right)= & jr^{H}G_{RF}^{H}V_{k}P_{b}^{⊥}{\left(G_{RF}^{H}\right)}^{†}r-jr^{H}G_{RF}^{H}P_{b}^{⊥}V_{k}{\left(G_{RF}^{H}\right)}^{†}r. \end{array}$$By substituting (Equation 22) into (Equation 25), $\dot{g}\left(v_{1},v_{2}|v_{k}\right)$ in (Equation 25) becomes
(26)$$\begin{array}{cc} \dot{g}\left(v_{1},v_{2}|v_{k}\right)= & j{\left(r_{s}+n\right)}^{H}G_{RF}^{H}V_{k}P_{b}^{⊥}{\left(G_{RF}^{H}\right)}^{†}\left(r_{s}+n\right)-j{\left(r_{s}+n\right)}^{H}G_{RF}^{H}P_{b}^{⊥}W_{e}{\left(G_{RF}^{H}\right)}^{†}\left(r_{s}+n\right)\\ = & -2ℑ\left\{r_{s}^{H}G_{RF}^{H}V_{k}P_{b}^{⊥}{\left(G_{RF}^{H}\right)}^{†}n\right\}-2ℑ\left\{n^{H}G_{RF}^{H}V_{k}P_{b}^{⊥}{\left(G_{RF}^{H}\right)}^{†}n\right\}. \end{array}$$Due to the independence between n and $r_{s}$, we have
(27)$$\begin{array}{cc} E\{\dot{g}\left(v_{1},v_{2}|v_{k}\right)\} & =-2\sigma_{n}^{2}ℑ\left\{\mathrm{tr}\left\{G_{RF}^{H}V_{k}P_{b}^{⊥}{\left(G_{RF}^{H}\right)}^{†}\right\}\right\}=0. \end{array}$$The second-order conditional derivative $\ddot{g}\left(v_{1},v_{2}|v_{k}\right)$ with respect to $v_{k}$ is written as
(28)$$\begin{array}{cc} \ddot{g}\left(v_{1},v_{2}|v_{k}\right) & =-r^{H}G_{RF}^{H}V_{k}^{2}P_{b}^{⊥}{\left(G_{RF}^{H}\right)}^{†}n+r^{H}G_{RF}^{H}V_{k}P_{b}^{⊥}V_{k}{\left(G_{RF}^{H}\right)}^{†}r\\ & +r^{H}G_{RF}^{H}V_{k}P_{b}^{⊥}V_{k}{\left(G_{RF}^{H}\right)}^{†}r-n^{H}G_{RF}^{H}P_{b}^{⊥}V_{k}^{2}{\left(G_{RF}^{H}\right)}^{†}r. \end{array}$$Similarly, $E\{\ddot{g}\left(v_{1},v_{2}|v_{k}\right)\}$ is calculated as
(29)$$\begin{array}{cc} E\{\ddot{g}\left(v_{1},v_{2}|v_{k}\right)\} & =2r_{s}^{H}G_{RF}^{H}V_{k}P_{b}^{⊥}V_{k}{\left(G_{RF}^{H}\right)}^{†}r_{s}+E\left\{\mathrm{tr}\left(-V_{k}^{2}P_{b}^{⊥}+2V_{k}P_{b}^{⊥}V_{k}-P_{b}^{⊥}V_{k}^{2}\right)\right\}\\ = & 2\sigma_{s}^{2}{\left|\alpha \right|}^{2}\mathrm{vec}{\left(A\right)}^{H}V_{k}P_{b}^{⊥}V_{k}\mathrm{vec}\left(A\right). \end{array}$$Based on (Equation 28) and (Equation 29), $\ddot{g}\left(v_{1},v_{2}|v_{k}\right)$ is further written as
(30)$$\begin{array}{c} \ddot{g}\left(v_{1},v_{2}|v_{k}\right)=E\left\{\ddot{g}\left(v_{1},v_{2}|v_{k}\right)\right\}+f_{2}\left(n\right)+f_{2}\left(n^{2}\right), \end{array}$$
where $f_{2}\left(n\right)$ and $f_{2}\left(n^{2}\right)$ represent the linear and quadrature functions of n in $\ddot{g}\left(v_{1},v_{2}|v_{k}\right)$, respectively. $\dot{g}\left(v_{1},v_{2}|v_{k}\right)$ can be written as
(31)$$\begin{array}{c} \dot{g}\left(v_{1},v_{2}|v_{k}\right)=f_{1}\left(n\right)+f_{1}\left(n^{2}\right), \end{array}$$
where $f_{1}\left(n\right)$ and $f_{1}\left(n^{2}\right)$ are the linear and quadrature functions of n in $\dot{g}\left(v_{1},v_{2}|v_{k}\right)$, respectively.Substituting (Equation 30) and (Equation 31) into (Equation 24), $\Delta v_{k}$ in (Equation 24) can be rewritten as
(32)$$\begin{array}{cc} \Delta v_{k} & =-\frac{f_{1}\left(n\right)+f_{1}\left(n^{2}\right)}{E\left\{\ddot{g}\left(v_{1},v_{2}|v_{k}\right)\right\}+f_{2}\left(n\right)+f_{2}\left(n^{2}\right)}\\ \; & =-\frac{f_{1}\left(n\right)+f_{1}\left(n^{2}\right)}{E\{\ddot{g}\left(v_{1},v_{2}|v_{k}\right)\}}\left(1-\frac{f_{2}\left(n\right)+f_{2}\left(n^{2}\right)}{E\{\ddot{g}\left(v_{1},v_{2}|v_{k}\right)\}}+{\left(\frac{f_{2}\left(n\right)+f_{2}\left(n^{2}\right)}{E\{\ddot{g}\left(v_{1},v_{2}|v_{k}\right)\}}\right)}^{2}-\cdots \right)\\ \; & \approx -\frac{f_{1}\left(n\right)+f_{1}\left(n^{2}\right)}{E\{\ddot{g}\left(v_{1},v_{2}|v_{k}\right)\}}=-\frac{\dot{g}\left(v_{1},v_{2}|v_{k}\right)}{E\{\ddot{g}\left(v_{1},v_{2}|v_{k}\right)\}}. \end{array}$$$c_{\Delta v_{k}}$ is expressed by
(33)$$\begin{array}{cc} c_{\Delta v_{k}} & =\frac{E\{\dot{g}{\left(v_{1},v_{2}|v_{k}\right)}^{2}\}}{E{\left\{\ddot{g}\left(v_{1},v_{2}|v_{k}\right)\right\}}^{2}}, \end{array}$$
where
(34)$$\begin{array}{ccc} E & \{\dot{g}{\left(v_{1},v_{2}|v_{k}\right)}^{2}\}\; & \approx 2\sigma_{n}^{2}r_{s}^{H}G_{RF}^{H}V_{k}P_{b}^{⊥}V_{k}{\left(G_{RF}^{H}\right)}^{†}r_{s}=2\sigma_{s}^{2}\sigma_{n}^{2}{\left|\alpha \right|}^{2}\mathrm{vec}{\left(A\right)}^{H}V_{k}P_{b}^{⊥}V_{k}\mathrm{vec}\left(A\right). \end{array}$$Substituting (Equation 27), (Equation 29) and (Equation 34) into (Equation 33) yields (Equation 19). □

Using the formulations of $v_{1}$ and $v_{2}$ in Section 2.2, ϕ and θ are written, respectively, as
(35)$$\begin{array}{c} \phi =\cos^{-1}\left(\frac{\lambda v_{1}}{2\pi d}\right), \end{array}$$
(36)$$\begin{array}{c} \theta =\cos^{-1}\left(\frac{\lambda v_{2}}{2\pi d\sin\phi }\right). \end{array}$$

Based on (Equation 19), (Equation 35) and (Equation 36), E{Δϕ}=E{Δθ}=0, and $E\{\Delta \phi^{2}\}$ is expressed as
(37)$$\begin{array}{cc} c_{\Delta \phi } & =E\left\{\left(\hat{\phi }-\phi \right){\left(\hat{\phi }-\phi \right)}^{H}\right\}=\frac{{\left(\frac{\lambda }{2\pi d}\right)}^{2}}{1-{\left(\frac{\lambda v_{1}}{2\pi d}\right)}^{2}}\frac{\sigma_{n}^{2}}{2\sigma_{s}^{2}{\left|\alpha \right|}^{2}\mathrm{vec}{\left(A\right)}^{H}V_{1}P_{b}^{⊥}V_{1}\mathrm{vec}\left(A\right)}, \end{array}$$
(38)$$\begin{array}{cc} c_{\Delta \theta } & =E\left\{\left(\hat{\theta }-\theta \right){\left(\hat{\theta }-\theta \right)}^{H}\right\}=\frac{{\left(\frac{\lambda }{2\pi d\sin\phi }\right)}^{2}}{1-{\left(\frac{\lambda v_{2}}{2\pi d\sin\phi }\right)}^{2}}\frac{\sigma_{n}^{2}}{2\sigma_{s}^{2}{\left|\alpha \right|}^{2}\mathrm{vec}{\left(A\right)}^{H}V_{2}P_{b}^{⊥}V_{2}\mathrm{vec}\left(A\right)}. \end{array}$$

Based on (Equation 17), $\hat{\alpha }$ is rewritten as
(39)$$\begin{array}{cc} \hat{\alpha } & =\mathrm{vec}{\left(\hat{A}\right)}^{H}{\left(G_{RF}^{H}\right)}^{†}s^{*}\left(G_{RF}^{H}\mathrm{vec}\left(A\right)\alpha s^{*}+n\right)\\ \; & =\sigma_{s}^{2}\mathrm{vec}{\left(\hat{A}\right)}^{H}\mathrm{vec}\left(A\right)\alpha +\mathrm{vec}{\left(\hat{A}\right)}^{H}{\left(G_{RF}^{H}\right)}^{†}s^{*}n, \end{array}$$
where $\mathrm{vec}{\left(\hat{A}\right)}^{H}$ is calculated by the estimation of $\left({\hat{v}}_{1},{\hat{v}}_{2}\right)$. By using Taylor’s expansion, we approximate $\mathrm{vec}{\left(\hat{A}\right)}^{H}$ as
(40)$$\begin{array}{c} \mathrm{vec}{\left(\hat{A}\right)}^{H}\approx \mathrm{vec}\left(A\right)+j\mathrm{vec}{\left(A\right)}^{H}V_{k}\Delta v_{k},\;k=1,2. \end{array}$$

Substituting (Equation 40) into (Equation 39), $\hat{\alpha }$ in (Equation 39) can be written as
(41)$$\begin{array}{cc} \hat{\alpha } & =\sigma_{s}^{2}\left(\mathrm{vec}{\left(A\right)}^{H}+j\mathrm{vec}{\left(A\right)}^{H}V_{1}\Delta v_{1}\right)\mathrm{vec}\left(A\right)\alpha +\mathrm{vec}{\left(\hat{A}\right)}^{H}{\left(G_{RF}^{H}\right)}^{†}s^{*}n\\ \; & =\alpha +j\mathrm{vec}{\left(A\right)}^{H}V_{1}\mathrm{vec}\left(A\right)\Delta v_{1}\alpha +\mathrm{vec}{\left(\hat{A}\right)}^{H}{\left(G_{RF}^{H}\right)}^{†}s^{*}n. \end{array}$$

Based on (Equation 19), the statistical characteristics of the estimation error of the mmWave path gain is calculated as
(42)$$\begin{array}{cc} \mu_{\Delta \alpha }=E\left\{\Delta \alpha \right\} & =E\{j\mathrm{vec}{\left(A\right)}^{H}V_{1}\mathrm{vec}\left(A\right)\Delta v_{1}\alpha +\mathrm{vec}{\left(\hat{A}\right)}^{H}{\left(G_{RF}^{H}\right)}^{†}s^{*}n\}=0, \end{array}$$
(43)$$\begin{array}{c} c_{\Delta \alpha }=E\left\{\left(\hat{\alpha }-\alpha \right){\left(\hat{\alpha }-\alpha \right)}^{H}\right\}=\frac{\sigma_{n}^{2}{\left|\mathrm{vec}{\left(A\right)}^{H}V_{1}\mathrm{vec}\left(A\right)\right|}^{2}}{2\sigma_{s}^{2}\mathrm{vec}{\left(A\right)}^{H}V_{1}P_{b}^{⊥}V_{1}\mathrm{vec}\left(A\right)}+\sigma_{n}^{2}. \end{array}$$

### 3.2. Location Validation

The location validation is implemented by comparing the similarity between the current channel parameters and the previous ones with preset thresholds, based on a binary hypothesis test. If we denote the difference between the current channel parameters (i.e., ϕ,α,θ) at successive time as $\Theta_{\zeta }$ for ζ∈{ϕ,α,θ}, then we have
(44)$$\begin{array}{c} \Theta_{\zeta }=\sum_{l=1}^{L}{\left|{\hat{\zeta }}_{l,XB}\left(t\right)-{\hat{\zeta }}_{l,AB}\left(t-1\right)\right|}^{2},\zeta \in \left\{\phi ,\alpha ,\theta \right\}. \end{array}$$

Let $\eta_{\phi }$, $\eta_{\theta }$ and $\eta_{\alpha }$ be preset thresholds for AAoA, EAoA and path gain, respectively. Next, we design a simple binary hypothesis testing for location validation as
(45)$$\left\{\begin{array}{c} H_{0}:\Theta_{\zeta }\leq \eta_{\zeta },\;\;\zeta \in \left\{\phi ,\alpha ,\theta \right\},\\ H_{1}:\Theta_{\zeta }>\eta_{\zeta },\;\;\zeta \in \left\{\phi ,\alpha ,\theta \right\}, \end{array}\right.$$

Under $H_{0}$, the current transmitter is Alice. In contrast, $H_{1}$ means that the current transmitter is Eve.

The fundamental principle for the proposed scheme in this paper is that location validation decision is implemented by comparing the similarities between current estimated channel features at time *t* and previous ones at time t−1 with the corresponding preset thresholds, based on the framework of hypothesis testing theory. Channel statistics and the temporal values for AAoA, EAoA and path gain at time t−1, and correlation coefficients can be modeled as side information that is known at both the transmitter (Alice) and the legitimate receiver (Bob), but not the adversary (Eve). The legitimate transmitter (Alice) and intended legitimate receiver (Bob) cooperate to secure communication transmission by means of the knowledge of this side information; this is beneficial for both partners. It should be stressed that the validation and efficiency of this idea were already studied from an information-theoretical point-of-view in [30], and the work [30] also provided the right guidance to the appropriate code design in practice.

## 4. Performance Analysis

In this section, we focus on the modeling of false alarm rate (denoted by $P_{f}$) and detection rate (denoted by $P_{d}$). To facilitate the calculation of the two probabilities, we denote $P_{\Theta_{\zeta },H_{i}}$ the probability of $\left(\Theta_{\zeta }>\eta_{\zeta }\right)$ under $H_{i}$ for i=1,2 and ζ∈{ϕ,α,θ}, and we consider $\sigma_{\zeta_{l,AB}}^{2}=\sigma_{\zeta_{AB}}^{2}$, and $\sigma_{\zeta_{l,EB}}^{2}=\sigma_{\zeta_{EB}}^{2}$. Based on authentication decision in (Equation 45), we establish the following lemmas regarding the probability $P_{\Theta_{\zeta },H_{i}}$.

**Lemma 2.** 
*Based on (Equation 44) and (Equation 45), $P_{\Theta_{\phi },H_{0}}$ is expressed by*

(46)
$$\begin{array}{c} P_{\Theta_{\phi },H_{i}}=\exp\left(-\frac{\eta_{\phi }}{2\sigma_{\phi ,H_{i}}^{2}}\right)\sum_{z=0}^{\frac{L}{2}-1}\frac{1}{z!}{\left(\frac{\eta_{\phi }}{2\sigma_{\phi ,H_{i}}^{2}}\right)}^{z}, \end{array}$$

*where $\sigma_{\phi ,H_{0}}^{2}=2\left(1-\rho_{\phi }\right)\sigma_{\phi_{AB}}^{2}+2\sigma_{\Delta_{\phi }}^{2}$ and $\sigma_{\phi ,H_{1}}^{2}=\sigma_{\phi_{EB}}^{2}+\sigma_{\phi_{AB}}^{2}+2\sigma_{\Delta_{\phi }}^{2}$.*


**Proof.** To calculate the probability distribution of $\Theta_{\phi }$ in (Equation 44), we define Λϕ as
(47)$$\begin{array}{c} \Lambda \phi ≜{\hat{\phi }}_{l,XB}\left(t\right)-{\hat{\phi }}_{l,AB}\left(t-1\right). \end{array}$$Under $H_{0}$, by substituting (Equation 9) into (Equation 47), $\Lambda \phi_{H_{0}}$ is obtained as
(48)$$\begin{array}{cc} \Lambda \phi_{H_{0}} & ={\hat{\phi }}_{l,AB}\left(t\right)-{\hat{\phi }}_{l,AB}\left(t-1\right)\\ \; & =\left(\rho_{\phi }-1\right)\phi_{l,AB}\left(t-1\right)+\sqrt{1-\rho_{\phi }^{2}}u_{\phi ,l,XB}\left(t\right)+\Delta \phi_{l}\left(t\right)-\Delta \phi_{l}\left(t-1\right). \end{array}$$One can see that $\Lambda \phi_{H_{0}}$ is a zero-mean Gaussian distribution with variance $\sigma_{\phi ,H_{0}}^{2}=2\left(1-\rho_{\phi }\right)\sigma_{\phi_{AB}}^{2}+2\sigma_{\Delta_{\phi }}^{2}$. Therefore, $\Theta_{\phi }$ under $H_{0}$ is a central chi-square distribution random variable and the PDF of $\Theta_{\phi }$ under $H_{0}$ is then written as
(49)$$\begin{array}{c} f_{\Theta_{\phi },H_{0}}\left(x\right)=\frac{{\left(\frac{x}{2\sigma_{\phi ,H_{0}}^{2}}\right)}^{\frac{L}{2}-1}}{2\sigma_{\phi ,H_{0}}^{2}\Gamma \left(L/2\right)}\exp\left(-\frac{x}{2\sigma_{\phi ,H_{0}}^{2}}\right),\;x\geq 0, \end{array}$$
where Γ(·) is a Gamma function. The CDF of $\Theta_{\phi }$ under $H_{0}$ is written as
(50)$$\begin{array}{c} F_{\Theta_{\phi },H_{0}}\left(x\right)=1-\exp\left(-\frac{x}{2\sigma_{\phi ,H_{0}}^{2}}\right)\sum_{z=0}^{\frac{L}{2}-1}\frac{1}{z!}{\left(\frac{x}{2\sigma_{\phi ,H_{0}}^{2}}\right)}^{z}. \end{array}$$Accordingly, $P_{\Theta_{\phi },H_{0}}$ is written as
(51)$$\begin{array}{cc} P_{\Theta_{\phi },H_{0}} & =\mathrm{Pr}\left\{\Theta_{\phi }>\eta_{\phi }|H_{0}\right\}=\exp\left(-\frac{\eta_{\phi }}{2\sigma_{\phi ,H_{0}}^{2}}\right)\sum_{z=0}^{\frac{L}{2}-1}\frac{1}{z!}{\left(\frac{\eta_{\phi }}{2\sigma_{\phi ,H_{0}}^{2}}\right)}^{z}. \end{array}$$Similarly, based on (Equation 47), we derive $\Lambda \phi_{H_{1}}$ under hypothesis $H_{1}$ as
(52)$$\begin{array}{cc} \Lambda \phi_{H_{1}} & ={\hat{\phi }}_{l,EB}\left(t\right)-{\hat{\phi }}_{l,AB}\left(t-1\right)=\phi_{l,EB}\left(t\right)+\Delta \phi_{l}\left(t\right)-\phi_{l,AB}\left(t-1\right)-\Delta \phi_{l}\left(t-1\right). \end{array}$$Because $\phi_{l,EB}\left(t\right)$, $\Delta \phi_{l}\left(t\right)$, $\phi_{l,AB}\left(t-1\right)$ and $\Delta \phi_{l}\left(t-1\right)$ are independent zero-mean Gaussian random variables with variances $\sigma_{\phi_{l,EB}}^{2}$, $\sigma_{\Delta_{\phi }}^{2}$, $\sigma_{\phi_{l,AB}}^{2}$ and $\sigma_{\Delta_{\phi }}^{2}$, respectively, and then the PDF of $\Theta_{\phi }$ under $H_{1}$ is written as
(53)$$\begin{array}{c} f_{\Theta_{\phi },H_{1}}\left(x\right)=\frac{{\left(\frac{x}{2\sigma_{\phi ,H_{1}}^{2}}\right)}^{\frac{L}{2}-1}}{2\sigma_{\phi ,H_{1}}^{2}\Gamma \left(L/2\right)}\exp\left(-\frac{x}{2\sigma_{\phi ,H_{1}}^{2}}\right),\;x\geq 0, \end{array}$$
where $\sigma_{\phi ,H_{1}}^{2}=\sigma_{\phi_{EB}}^{2}+\sigma_{\phi_{AB}}^{2}+2\sigma_{\Delta_{\phi }}^{2}$. Then, the CDF of $\Theta_{\phi }$ under $H_{1}$ is written as
(54)$$\begin{array}{c} F_{\Theta_{\phi },H_{1}}\left(x\right)=1-\exp\left(-\frac{x}{2\sigma_{\phi ,H_{1}}^{2}}\right)\sum_{z=0}^{\frac{L}{2}-1}\frac{1}{z!}{\left(\frac{x}{2\sigma_{\phi ,H_{1}}^{2}}\right)}^{z}. \end{array}$$According to (Equation 54), $P_{\Theta_{\phi },H_{1}}$ is written as
(55)$$\begin{array}{cc} P_{\Theta_{\phi },H_{1}} & =\mathrm{Pr}\left\{\Theta_{\phi }>\eta_{\phi }|H_{1}\right\}=\exp\left(-\frac{\eta_{\phi }}{2\sigma_{\phi ,H_{1}}^{2}}\right)\sum_{z=0}^{\frac{L}{2}-1}\frac{1}{z!}{\left(\frac{\eta_{\phi }}{2\sigma_{\phi ,H_{1}}^{2}}\right)}^{z}. \end{array}$$□

**Lemma 3.** *Using* (Equation 45)*, $P_{\Theta_{\theta },H_{i}}$ (i=1,2) is evaluated as*
(56)$$\begin{array}{c} P_{\Theta_{\theta },H_{i}}=\exp\left(-\frac{\eta_{\theta }}{2\sigma_{\theta ,H_{i}}^{2}}\right)\sum_{z=0}^{\frac{L}{2}-1}\frac{1}{z!}{\left(\frac{\eta_{\theta }}{2\sigma_{\theta ,H_{i}}^{2}}\right)}^{z}, \end{array}$$
*where $\sigma_{\theta ,H_{0}}^{2}=2\left(1-\rho_{\theta }\right)\sigma_{\theta_{AB}}^{2}+2\sigma_{\Delta_{\theta }}^{2}$ and $\sigma_{\theta ,H_{1}}^{2}=\sigma_{\theta_{EB}}^{2}+\sigma_{\theta_{AB}}^{2}+2\sigma_{\Delta_{\theta }}^{2}$.*

**Lemma 4.** *Using* (Equation 45)*, $P_{\Theta_{\alpha },H_{i}}$ (i=1,2) is evaluated as*
(57)$$\begin{array}{c} P_{\Theta_{\alpha },H_{i}}=\exp\left(-\frac{\eta_{\alpha }}{2\sigma_{\alpha ,H_{i}}^{2}}\right)\sum_{z=0}^{L-1}\frac{1}{z!}{\left(\frac{\eta_{\alpha }}{2\sigma_{\alpha ,H_{i}}^{2}}\right)}^{z}, \end{array}$$
*where $\sigma_{\alpha ,H_{0}}^{2}=2\left(1-\rho_{\alpha }\right)\sigma_{\alpha_{AB}}^{2}+2\sigma_{\Delta_{\alpha }}^{2}$ and $\sigma_{\alpha ,H_{1}}^{2}=\sigma_{\alpha_{EB}}^{2}+\sigma_{\alpha_{AB}}^{2}+2\sigma_{\Delta_{\alpha }}^{2}$.*

Based on the above lemmas, we can establish the following theorem regarding the closed expressions of $P_{f}$ and $P_{d}$.

**Theorem 1.** 
*For the mmWave MIMO communication model shown in Figure 1, $P_{f}$ and $P_{d}$ of the proposed authentication scheme are determined, respectively, as*

(58)
$$\begin{array}{c} P_{f}=P_{\Theta_{\phi ,H_{0}}}P_{\Theta_{\theta ,H_{0}}}P_{\Theta_{\alpha ,H_{0}}}, \end{array}$$


(59)
$$\begin{array}{c} P_{d}=1-\left(1-P_{\Theta_{\phi ,H_{1}}}\right)\left(1-P_{\Theta_{\theta ,H_{1}}}\right)\left(1-P_{\Theta_{\alpha ,H_{1}}}\right). \end{array}$$



Suppose that Bob has no knowledge of the channel parameters from data frame at time *t* such as ${\hat{\zeta }}_{l,AB}\left(t-1\right)$ and $\rho_{\zeta }$ for ζ∈{ϕ,α,θ}, l=1,...,L. In this case $\Theta_{\zeta }$ becomes
(60)$$\begin{array}{c} \Theta_{\zeta }=\sum_{l=1}^{L}{\left|{\hat{\zeta }}_{l,XB}\left(t\right)\right|}^{2},\zeta \in \left\{\phi ,\alpha ,\theta \right\}. \end{array}$$

Following similar steps for the derivation of $P_{\Theta_{\zeta },H_{i}}$, we can obtain $P_{\Theta_{\zeta },H_{i}}^{u}$ with *u* being without the knowledge of channel parameters at time t−1, and then we have
(61)$$\begin{array}{c} P_{\Theta_{\phi },H_{i}}^{u}=\exp\left(-\frac{\eta_{\phi }}{2\sigma_{\phi ,H_{i}}^{2}}\right)\sum_{z=0}^{\frac{L}{2}-1}\frac{1}{z!}{\left(\frac{\eta_{\phi }}{2\sigma_{\phi ,H_{i}}^{2}}\right)}^{z}, \end{array}$$
(62)$$\begin{array}{c} P_{\Theta_{\theta },H_{i}}^{u}=\exp\left(-\frac{\eta_{\theta }}{2\sigma_{\theta ,H_{i}}^{2}}\right)\sum_{z=0}^{\frac{L}{2}-1}\frac{1}{z!}{\left(\frac{\eta_{\theta }}{2\sigma_{\theta ,H_{i}}^{2}}\right)}^{z}, \end{array}$$
(63)$$\begin{array}{c} P_{\Theta_{\alpha },H_{i}}^{u}=\exp\left(-\frac{\eta_{\alpha }}{2\sigma_{\alpha ,H_{i}}^{2}}\right)\sum_{z=0}^{L-1}\frac{1}{z!}{\left(\frac{\eta_{\alpha }}{2\sigma_{\alpha ,H_{i}}^{2}}\right)}^{z}, \end{array}$$
where $\sigma_{\phi ,H_{0}}^{2}=\sigma_{\phi_{AB}}^{2}+\sigma_{\Delta_{\phi }}^{2}$, $\sigma_{\phi ,H_{1}}^{2}=\sigma_{\phi_{EB}}^{2}+\sigma_{\Delta_{\phi }}^{2}$, $\sigma_{\theta ,H_{0}}^{2}=\sigma_{\theta_{AB}}^{2}+\sigma_{\Delta_{\theta }}^{2}$, $\sigma_{\theta ,H_{1}}^{2}=\sigma_{\theta_{EB}}^{2}+\sigma_{\Delta_{\theta }}^{2}$, $\sigma_{\alpha ,H_{0}}^{2}=\sigma_{\alpha_{AB}}^{2}+\sigma_{\Delta_{\alpha }}^{2}$, and $\sigma_{\alpha ,H_{1}}^{2}=\sigma_{\alpha_{EB}}^{2}+\sigma_{\Delta_{\alpha }}^{2}$. In this case, $P_{f}^{u}$ and $P_{d}^{u}$ of the proposed authentication scheme are determined, respectively, as
(64)$$\begin{array}{c} P_{f}^{u}=P_{\Theta_{\phi ,H_{0}}}^{u}P_{\Theta_{\theta ,H_{0}}}^{u}P_{\Theta_{\alpha ,H_{0}}}^{u}, \end{array}$$
(65)$$\begin{array}{c} P_{d}^{u}=1-\left(1-P_{\Theta_{\phi ,H_{1}}}^{u}\right)\left(1-P_{\Theta_{\theta ,H_{1}}}^{u}\right)\left(1-P_{\Theta_{\alpha ,H_{1}}}^{u}\right). \end{array}$$

Suppose that legitimate Alice and adversary Eve transmit signals to Bob with equal probability. We use $P_{s}$ to denote the secrecy rate improvement afforded by the knowledge of data frame at time t−1 for both legitimate transmitter and receiver, relative to the case in which the knowledge is not used. Then, $P_{s}$ can be calculated as
(66)$$\begin{array}{c} P_{s}=\frac{1}{2}\left(P_{f}^{u}-P_{f}+P_{d}-P_{d}^{u}\right) \end{array}$$

## 5. Numerical Results

### 5.1. System Parameters and Simulation Settings

Let signal-to-noise ratio (SNR) denote as $\mathrm{SNR}=\frac{p}{\sigma_{n}^{2}}$ and $k_{\phi }=\frac{\sigma_{\phi_{l,EB}}^{2}}{\sigma_{\phi_{l,AB}}^{2}}$ be the ratio of Eve’s and Alice’s AAoA variances. We use $k_{\theta }=\frac{\sigma_{\theta_{l,EB}}^{2}}{\sigma_{\theta_{l,AB}}^{2}}$ to denote the ratio of Eve’s and Alice’s EAoA variances, and $k_{\alpha }=\frac{\sigma_{\alpha_{l,EB}}^{2}}{\sigma_{\alpha_{l,AB}}^{2}}$ to denote the ratio of Eve’s and Alice’s path gain variances. We set the number of the receiver antennas as $N_{r}=M\times N$. In the simulations, the first frame is validated based on a higher layer verification scheme and the second frame is validated by using the proposed location verification scheme.

### 5.2. Impact of SNR on the Verification Performance

We plot in Figure 2 ROC (receiver operating characteristic, ROC) variations under the settings of ($N_{r}$ = 128, *L* = 20, $k_{\phi }$ = 2 dB, $k_{\theta }$ = 2 dB, $k_{\alpha }$ = 2 dB) at different SNRs. From Figure 2, we can see that the authentication performance ($P_{f}$ and $P_{d}$) under SNR = 5 dB outperforms that of the others and the performance for SNR = −5 dB is poorest. This indicates that the SNRs have severe impacts on the authentication performance. One can improve the authentication performance by setting a higher SNR. However, the SNR level usually has a limit due to the practical conditions. In many practical environments, $P_{f}<0.1$ and $P_{d}>0.9$ are usually a required constraint pair to satisfy the requirements of a practical authentication system. As shown in Figure 2, the authentication performance ($P_{f}$ and $P_{d}$) for SNR = 0 dB and SNR = 5 dB can satisfy the practical requirements and the performance at SNR = 5 dB is best. From Figure 2, we also can see that when $P_{f}$ is fixed, $P_{d}$ increases when SNR increases. This means that a higher SNR can lead to a better authentication performance.

### 5.3. Impact of the Number of Antennas on the Authentication Performance

We show in Figure 3 the ROC results under different receiver antenna numbers with the settings of (SNR = 10 dB, *L* = 20, $k_{\phi }$ = 2 dB, $k_{\theta }$ = 2 dB, $k_{\alpha }$ = 2 dB). From Figure 3, we can see that the authentication performance ($P_{f}$, $P_{d}$) under 64 receiver antennas outperforms that of the others, and the performance under 16 receiver antennas is poorest. This is because multiple antennas can improve feature dimensions and further improve the authentication performance. When $P_{f}$ is fixed, $P_{d}$ increases when the number of receiver antennas increases. Moreover, the authentication performance ($P_{f}$, $P_{d}$) under 64 receiver antennas can satisfy the practical authentication requirements.

### 5.4. Impact of the Number of Multipaths on the Authentication Performance

We show in Figure 4 the ROC curves under different numbers of multipaths with the settings of (SNR = 10 dB, $N_{r}$ = 64, $k_{\phi }$ = 2 dB, $k_{\theta }$ = 2 dB, $k_{\alpha }$ = 2 dB). From Figure 4, we can see that the authentication performance ($P_{f}$, $P_{d}$) under *L* = 15 multipaths outperforms that of the others, and the performance under *L* = 5 multipaths is poorest. This means that more multipaths can contribute to the authentication performance. We can also see that when $P_{f}$ is fixed, $P_{d}$ increases when the number of multipaths increases.

### 5.5. Impact of $k_{\alpha }$ on the Authentication Performance

To investigate the impact of the distance between Eve–Bob and Alive–Bob on performance, we need to characterize $\sigma_{\alpha_{l,XB}}^{2}$ being the variance of the path gain of the *l*-th propagation path from the transmitter *X* to Bob. According to [31], baseband path gain $\alpha_{l,XB}$ is written as $\alpha_{l,XB}=\beta_{l,XB}\exp\left(-\frac{j2\pi f_{c}d_{l,XB}}{c}\right)$, where $\beta_{l,XB}$ is the attenuation coefficient of *l*-th propagation path and $f_{c}$ is carrier frequency. According to the Central Limit Theorem, the path gain can be modeled as a zero-mean complex circular symmetric Gaussian process and for large-scale fading channels, $\sigma_{\alpha_{l,XB}}^{2}$ can be modeled by applying ([32], Chapter 2), as
(67)$$\begin{array}{c} \sigma_{\alpha_{l,XB}}^{2}=K{\left(\frac{d_{0}}{d_{l,XB}}\right)}^{\varsigma }\Omega , \end{array}$$
where *K* is a reference path gain value; $d_{0}$ is a reference distance for antenna far-field; ς is a path loss exponent ranging between 2 and 5 in free space propagation wireless environments; and Ω is a shadowing factor modeled as a log-normal random variable. Because the distance between the transmitter and receiver is much larger than antenna separation, we can use the approximation $d_{l,XB}\approx \cdots \approx d_{L,XB}=d_{XB}$ for X={A,E}. From (Equation 67), we have $\sigma_{\alpha_{1,XB}}^{2}\approx \cdots \approx \sigma_{\alpha_{L,XB}}^{2}=\sigma_{\alpha_{XB}}^{2}$. Both Alice and Eve are assumed to be randomly deployed at arbitrary locations in a circular area centered on Bob, and there is no shadow fading in that area. By using (Equation 67), $k_{\alpha }$ by dB value without shadow fading is given by $k_{\alpha }=10\varsigma \log\left(\frac{d_{AB}}{d_{EB}}\right)$. We make an assumption (without loss of generality) that Alice is in a fixed location, i.e., $d_{AB}$ is a constant and we adjust the distance between Eve and Bob $d_{EB}$ to obtain different $k_{\alpha }$.

We show in Figure 5 how $P_{d}$ varies with $k_{\alpha }$ under different $P_{f}$s and with the settings of (SNR = 10 dB, $N_{r}$ = 128, *L* = 25, $k_{\phi }$ = 2 dB, $k_{\theta }$ = 2 dB). From Figure 5, $P_{d}$ increases when $k_{\alpha }$ increases for a given $P_{f}$. This clearly indicates that if Eve is closer to Bob, she might be successfully detected by Bob, and Eve might select an appropriate location in which she has a high probability of succesfully impersonating Alice. In addition, the authentication performance under $P_{f}$ = 0.08 outperforms that of the others, and the performance under $P_{f}$ = 0.02 is poorest. This is because a lower $P_{f}$ value means a stricter performance requirement and there would be a trade-off between $P_{d}$ and $P_{f}$. We can also see from Figure 5 that when $k_{\alpha }$ is fixed, $P_{d}$ increases when $P_{f}$ increases.

### 5.6. Performance Comparison

Figure 6 displays a performance comparison between the authentication scheme proposed in [16] and our scheme. From Figure 6, we can see that the performance ($P_{f}$, $P_{d}$) of our scheme jointly using AAoA, EAoA and path gain outperforms that of the scheme only using path gain [16]. This is because the scheme in [16] uses coarse-grained channel information while our scheme uses multi-dimensional mmWave MIMO channel features (i.e., AAoA, EAoA and path gain) to effectively utilize the fine-grained channel information. It is also noted that when $P_{f}$ increases, the detection rates of both schemes increase. Moreover, the authentication performance ($P_{f}$, $P_{d}$) of our scheme can better satisfy the practical authentication requirements. This is beneficial for exploiting both the AAoA and EAoA features, and allows us to gain important insights on the impacts of angle domain features on authentication performance.

### 5.7. Impact of $\rho_{\alpha }$ on the Authentication Performance

We show in Figure 7 ROC variations under the settings of (SNR = 10 dB, $N_{r}$ = 128, *L* = 20, $k_{\phi }$ = 2 dB, $k_{\theta }$ = 2 dB, $k_{\alpha }$ = 2 dB) at different correlation coefficients of path gain. From Figure 7, we can see that the authentication performance ($P_{f}$, $P_{d}$) under $\rho_{\alpha }$ = 0.9 outperforms that of the others, and the performance under $\rho_{\alpha }$ = 0.7 is the poorest. This is because a larger $\rho_{\alpha }$ means that the communication entities move slower. When $P_{f}$ is fixed, $P_{d}$ increases when $\rho_{\alpha }$ increases. Moreover, the authentication performance ($P_{f}$, $P_{d}$) under $\rho_{\alpha }$ = 0.9 can satisfy the practical authentication requirements.

## 6. Conclusions

By utilizing multi-dimensional mmWave MIMO channel features, this paper developed a location verification scheme for mmWave MIMO communication systems. Furthermore, we modeled the three-dimensional mmWave channel features (i.e., AAoA, EAoA and path gain) and extracted these feature parameters through maximum-likelihood estimator. We also derived the analytical expressions for the typical performance metrics. Numerical results indicate that the proposed authentication scheme can benefit from using multi-dimensional mmWave channel features. The results in this paper reveal that the proposed scheme is general in nature and can be applied to more communication systems, and we expect the methodology developed in this paper to be valuable for devising new physical layer authentication schemes in other network scenarios.

## Figures and Tables

**Figure 1 sensors-22-09202-f001:**
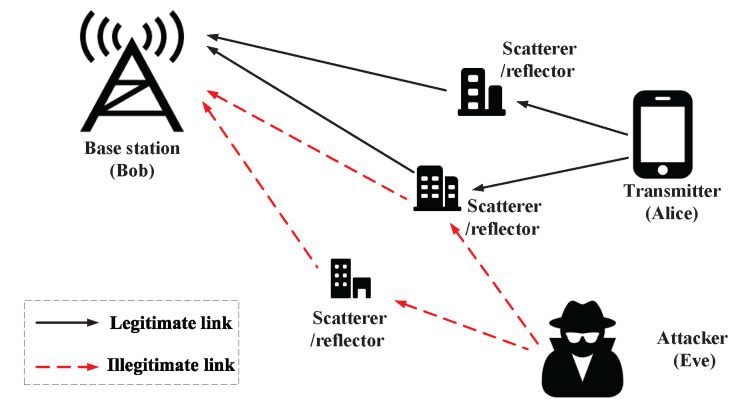
System model.

**Figure 2 sensors-22-09202-f002:**
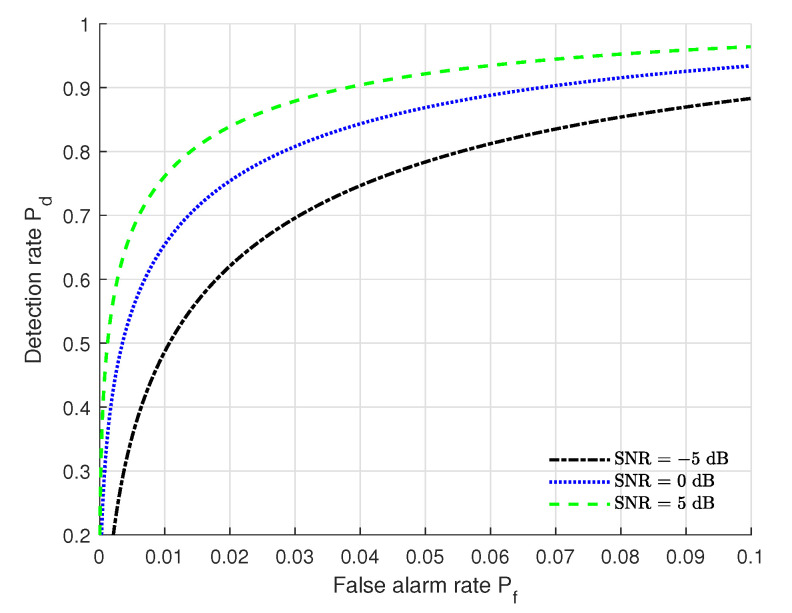
ROC curves under different SNRs with the settings of ($N_{r}$ = 128, *L* = 20, $k_{\phi }$ = 2 dB, $k_{\theta }$ = 2 dB, $k_{\alpha }$ = 2 dB).

**Figure 3 sensors-22-09202-f003:**
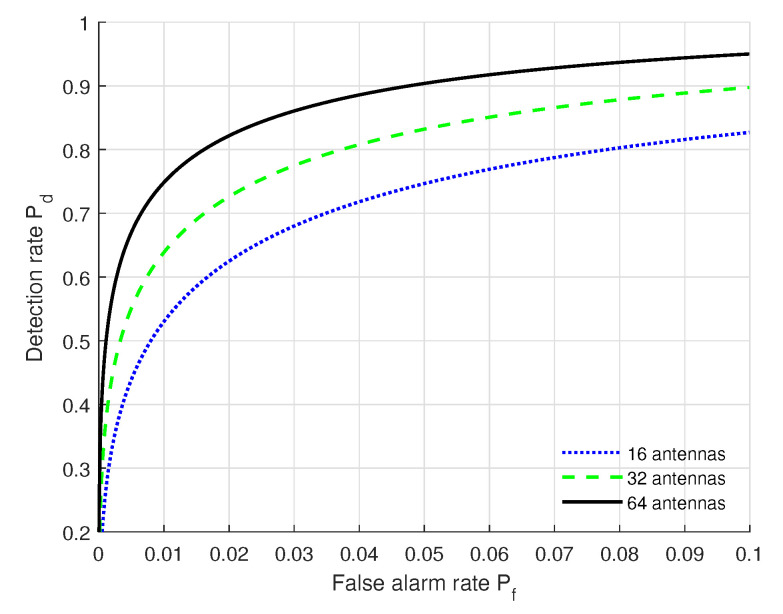
ROC curves under different numbers of receiver antennas with the settings of (SNR = 10 dB, *L* = 20, $k_{\phi }$ = 2 dB, $k_{\theta }$ = 2 dB, $k_{\alpha }$ = 2 dB).

**Figure 4 sensors-22-09202-f004:**
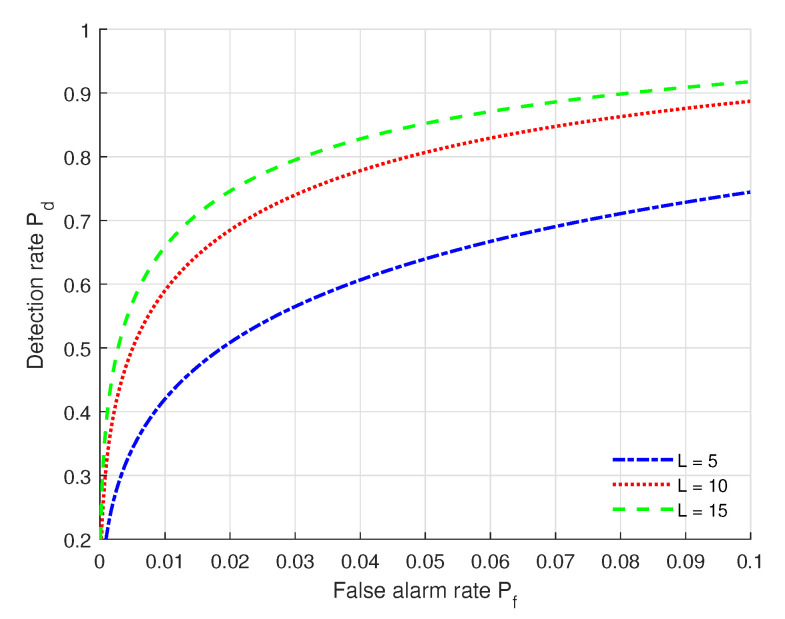
ROC curves under different numbers of multipaths with the settings of (SNR = 10 dB, $N_{r}$ = 64, $k_{\phi }$ = 2 dB, $k_{\theta }$ = 2 dB, $k_{\alpha }$ = 2 dB).

**Figure 5 sensors-22-09202-f005:**
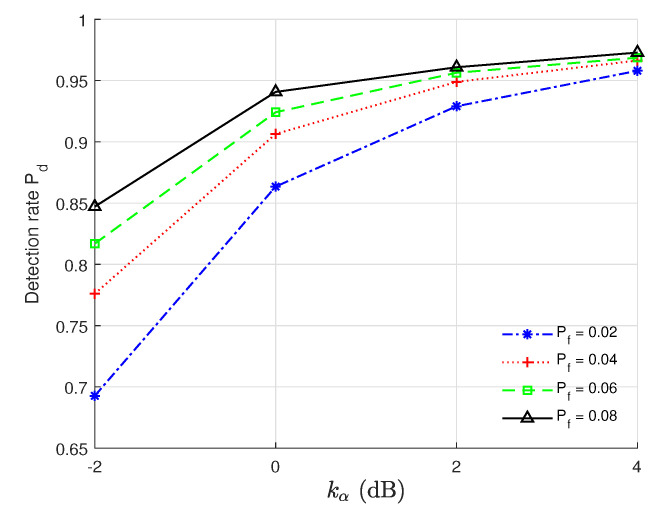
$P_{d}$ vs. $k_{\alpha }$ with the settings of (SNR = 10 dB, $N_{r}$ = 128, *L* = 25, $k_{\phi }$ = 2 dB, $k_{\theta }$ = 2 dB).

**Figure 6 sensors-22-09202-f006:**
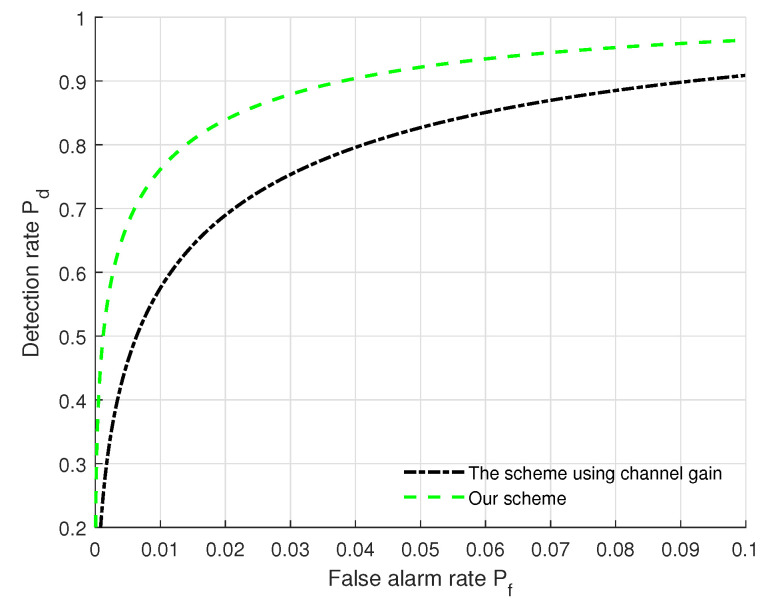
Performance comparison between the scheme in [16] and our scheme with the settings of (SNR = 5 dB, $N_{r}$ = 128, *L* = 20, $k_{\phi }$ = 2 dB, $k_{\theta }$ = 2 dB, $k_{\alpha }$ = 2 dB).

**Figure 7 sensors-22-09202-f007:**
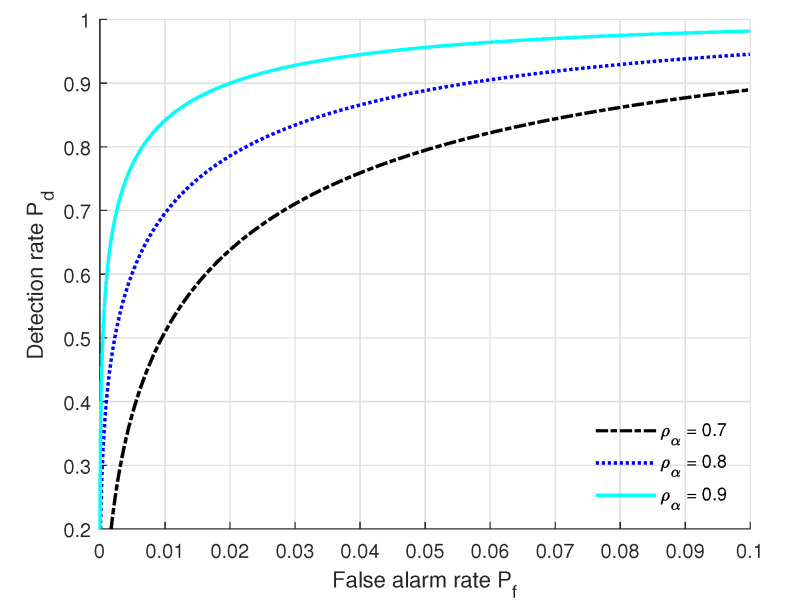
ROC curves under different correlation coefficients of path gain with the settings of (SNR = 10 dB, $N_{r}$ = 128, *L* = 20, $k_{\phi }$ = 2 dB, $k_{\theta }$ = 2 dB, $k_{\alpha }$ = 2 dB).

## Data Availability

Not applicable.

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
