# Peer review of "Utilizing Multi-Dimensional MmWave MIMO Channel Features for Location Verification"

_sensors, 2022, doi:10.3390/s22239202_

Round 1
Reviewer 1 Report
Comments to the authors:
1. Motivation and main difference between this paper should be presented more clearly. Also, difference with references [16]-[18] should be discussed.
2. References [7],[8],[10], [12]-[14] did not introduce in details in this paper.
3. New points/techniques in performance evaluation should be highlighted.
4. How is performance of the proposed method, as compared with the existing ones?
5. More discussion for figures should be added.
6. Typos and errors in this paper: signal[22] should be corrected by “signal [22].”, White space at page 11, etc.
Reviewer 2 Report
The results and methodology are interesting. I have few comments that I would like to see addressed before possibly recommending this paper for publication.
- The central idea is to use a hypothesis test to depart, at time frame [t], the legitimate transmitter (Alice) from the eavesdropper (Eve). Can you quantify the boost in secrecy rate afforded by the knowledge of data frame [t-1] at both transmitter and legitimate receiver, relative to the case in which that knowledge is not used.
- For the transmission at frame time [t] the data frame at time [t-1], which is shared with Bob, can be modeled as side-information that is known at both transmitter (Alice) and the legitimate receiver (Bob), but not the eavesdropper. The knowledge of this side information facilitates secure transmission during time [t], and how best this can be done information-theoretically was already studied and shown in
- Zaidi, and L. Vandendorpe, “Coding schemes for relay-assisted information embedding”, IEEE Transactions on Information Forensics and Security, vol. 4, no. 1, pp. 70-85, Jan. 2009.
Which actually has studied a more general model. The authors should elaborate on this connection and make a reference.
Reviewer 3 Report
The title and the abstract of the paper are misleading. The article does not address physical layer authentication using MIMO channels, while it just investigates the problem associated with location based authentication. The source (Alice) is not fingerprinted: authors address a different problem of authenticating the location of Alice over time by propagating the (supposed) authentication at time t-1 to time slot t.
Physical layer fingerprinting is a different topic involving the physical identification of the radio transmitter by resorting to unique peculiarities of the radio itself—under the assumption that two identical radios do not exist.
Nevertheless, the article has other critical issues. One of the most important parameter is the distance between Eve and Alice in order to measure the effectiveness of the proposed solution. Authors assume Eve being far away from Alice, then resorting to AAoA, EOA and gain to discriminate between the two sources at Bob's side. This is easy, maybe achievable without mmWave technology, and in some cases, without the MIMO architecture at Bob’s side—assuming Eve is in a completely different position from Alice.
Another problem is the scenario description. Location authentication works if and only if Alice and Bob stand still, indeed, if they move, the proposed solution does not work. This is not mentioned anywhere.
Finally, there are many typos while Fig. 1 is inconsistent with the text (Alice, Bob and Eve are missing).
Round 2
Reviewer 1 Report
This paper can be accepted for the publication.
Author Response
Thank you very much for your kind comments!
Reviewer 2 Report
The authors have revised the manuscript in a rather satisfactory manner; and I recommend Accept.
Author Response

(The authors gave the same response as above.)

Reviewer 3 Report
The paper has significantly improved during the first round of review but there is still a fundamental issue. This is not physical layer authentication but location verification as also acknowledged by the authors in the response letter. The title, the abstract and the introduction should be changed accordingly.
The second major issue is not fixed yet. The distance between Eve and Alive is an important parameter that should be addressed. This requires a clear evaluation in order to find the bounds according to which Eve breaks the location verification scheme.
Author Response
Response to Reviewers’ Comments
Manuscript ID: sensors-2011169Title: Physical Layer Authentication Using Multi-dimensional MmWave MIMO Channel Features
Dear Editor / Reviewers
Thank you very much for your efforts in handing/reviewing this paper! The detailed comments have helped us a lot in improving both the quality and the clarity of the paper.
In this revision, we have substantially revised the paper according to the reviewers’ comments. For each comment received, we have prepared a point-to-point response in the attached report. Changes in the revised manuscript are highlighted using different colors, with each color for one reviewer’s concerns.
Best Regards,
Sincerely Yours

Round 3
Reviewer 3 Report
Authors addressed all my concerns and the current version of the paper is ready for publication.